# Systematic Investigation of the Multifaceted Role of *SOX11* in Cancer

**DOI:** 10.3390/cancers14246103

**Published:** 2022-12-12

**Authors:** Qingqing Sun, Jun Du, Jie Dong, Shuaikang Pan, Hongwei Jin, Xinghua Han, Jinguo Zhang

**Affiliations:** 1Department of Medical Oncology, Anhui Provincial Hospital Affiliated to Anhui Medical University, Hefei 230001, China; 2Department of Medical Oncology, The First Affiliated Hospital of USTC, Division of Life Sciences and Medicine, University of Science and Technology of China, Hefei 230001, China; 3Department of Pathology, The First Affiliated Hospital of USTC, Division of Life Sciences and Medicine, University of Science and Technology of China, Hefei 230001, China

**Keywords:** pan-cancer, *SOX11*, prognosis, stemness, multi-analyses

## Abstract

**Simple Summary:**

The transcription factor *SOX11* is a member of the *SRY* box-containing family. Recently, *SOX11* research has shifted from embryogenesis to cancer development. Nevertheless, the roles of *SOX11* in different cancer types remain controversial. Here, we systematically explored the characteristics of *SOX11* by multiple omics analyses. Aberrant *SOX11* expression has been observed in many types of human malignant tumors. Moreover, *SOX11* expression could serve as an indicator of survival outcomes, tumor progression and cellular immune infiltration. Our work highlights the diverse role of *SOX11* in different types of tumors, which points out several directions for future prospective studies on *SOX11* in cancer.

**Abstract:**

SRY-box transcription factor 11 (*SOX11)*, as a member of the *SOX* family, is a transcription factor involved in the regulation of specific biological processes and has recently been found to be a prognostic marker for certain cancers. However, the roles of *SOX11* in cancer remain controversial. Our study aimed to explore the various aspects of *SOX11* in pan-cancer. The expression of *SOX11* was investigated by the Genotype Tissue-Expression (GTEX) dataset and the Cancer Genome Atlas (TCGA) database. The protein level of *SOX11* in tumor tissues and tumor-adjacent tissues was verified by human pan-cancer tissue microarray. Additionally, we used TCGA pan-cancer data to analyze the correlations among *SOX11* expression and survival outcomes, clinical features, stemness, microsatellite instability (MSI), tumor mutation burden (TMB), mismatch repair (MMR) related genes and the tumor immune microenvironment. Furthermore, the cBioPortal database was applied to investigate the gene alterations of *SOX11*. The main biological processes of *SOX11* in cancers were analyzed by Gene Set Enrichment Analysis (GSEA). As a result, aberrant expression of *SOX11* has been implicated in 27 kinds of cancer types. Aberrant *SOX11* expression was closely associated with survival outcomes, stage, tumor recurrence, MSI, TMB and MMR-related genes. In addition, the most frequent alteration of the *SOX11* genome was mutation. Our study also showed the correlations of *SOX11* with the level of immune infiltration in various cancers. In summary, our findings underline the multifaceted role and prognostic value of *SOX11* in pan-cancer.

## 1. Introduction

While new treatment strategies have been developed, cancer is still a significant cause of mortality worldwide. According to statistics, in China, an estimated 4,820,000 new cancer cases and 3,210,000 cancer deaths occurred in 2022 [1]. Molecular biomarkers, which can be sensitive enough to detect initial malignant transformations, have always been a prime focus of cancer research. The SRY-related HMG-box 11 transcription factor is one of the members of the *SOX* gene family that possess a highly conserved high mobility group (HMG) sequence [2]. According to the degree of homology of their domains, *SOX* genes in humans are classified into eight groups [3]. Together with *SOX4* and *SOX12*, *SOX11* is a subgroup of subgroup C [4], located on chromosome 2p25.3 [5]. They share a well-conserved C-terminal region [6]. *SOX11* regulates progenitor and stem cell behavior and often acts in regulating developmental processes, including neurogenesis and skeletogenesis [7].

*SOX11* is a protein-coding gene that plays a role in transcriptional regulation after forming a protein complex with other proteins [8,9]. Recently, various novel epigenetic regulatory mechanisms of *SOX11* have also been revealed [10,11]. *SOX11* is primarily expressed in various tissues of humans and mice, including the nervous system, gastrointestinal tract, lung, spleen, pancreas, kidneys and gonads [12,13,14]. Research on *SOX11* has shifted from its role in embryogenesis to disease development. In particular, the function of *SOX11* in carcinogenesis has become a major focus of interest. In many cancers, dysfunctional expression of *SOX11* has been correlated with increased cancer cell survival, inhibition of cell differentiation and tumor progression [15]. Nevertheless, in a limited number of malignancies, *SOX11* has also been identified to function as a tumor suppressor [16].

Previous studies have shown that the expression levels of *SOX11* vary in different cancer types. For example, a high level of *SOX11* mRNA can be detected in many types of tumors, including breast cancer [17], mantle cell lymphoma [18], epithelial ovarian cancer [19] and hepatocellular carcinoma [20]. In contrast, a low expression of *SOX11* was frequently observed in other cancers, such as nasopharyngeal carcinoma (NPC) [21], prostate cancer [22], gastric cancer (GC) [23] and bladder cancer [24]. Moreover, *SOX11* plays diverse roles in a variety of types of cancer. *SOX11* has been described as a key oncogenic factor in mantle cell lymphoma (MCL) [25]. A high level of *SOX11* expression is associated with poor overall survival (OS) and increased formation of metastasis in breast cancer patients [26,27]. *SOX11* can promote invasive transition in vivo, confirming its role in promoting the progression of DCIS to invasive breast cancer [28]. However, *SOX11* has been reported to act as a potential tumor suppressor that is down-regulated in prostate cancer and overexpression of *SOX11* suppresses migration and invasion in prostate cancer cells in vitro [22]. To date, studies on *SOX11* are limited by a single cancer type. Therefore, a systematic investigation the role of *SOX11* in cancer is required.

To reveal the role of *SOX11* in pan-cancer, we systematically integrated multiple databases from the perspective of bioinformatics. In this study, we comprehensively explored the expression of *SOX11* in normal tissues and its corresponding cancer tissues using the Genotype-Tissue Expression (GTEX) and The Cancer Genome Atlas (TCGA) dataset. A tissue microarray was used to verify the difference in *SOX11* expression among normal tissues, tumor tissues and corresponding tumor-adjacent tissues. The prognostic value of *SOX11* in predicting survival outcomes was also evaluated. Then, the potential relationships between *SOX11* expression and clinicopathological phenotypes, stemness score, TMB, MSI, MMR-related genes and tumor immune microenvironment were further explored. In addition, GSEA was used to clarify the biological function of *SOX11* in pan-cancer. Overall, this study highlights the multifaceted role of *SOX11* in a variety of cancers, which may provide a theoretical basis for using *SOX11* as a prognostic biomarker in cancer.

## 2. Materials and Methods

### 2.1. Data Processing and SOX11 Expression Analysis

Transcriptome data and clinical features of TCGA pan-cancer were downloaded from the UCSC XENA (https://xena.ucsc.edu/, accessed on 6 August 2022). Gene expression data for 31 normal human tissues were obtained from the GTEX website (https://www.gtexportal.org/, accessed on 6 August 2022). The mRNA expression of *SOX11* was extracted using Perl scripts (Version 5.30.0.1, http://strawberryperl.com/, accessed on 6 August 2022). The Wilcoxon rank-sum test was used to examine differences in *SOX11* expression. All statistical analyses and plotting of the results were performed using R version 4.0.2 software (https://www.Rproject.org, accessed on 10 July 2022).

### 2.2. Tissue Microarray and Human Protein Atlas (HPA) Database

Human pan-cancer tissue arrays (Cat# HOrgC120PG04, Shanghai Outdo Biotech Co., Ltd., Shanghai, China) were analyzed for the protein expression of *SOX11* by immunohistochemistry. Research ethics approval was granted by the Ethics Committee of Shanghai Outdo Biotech Co., Ltd. (protocol code SHYJS-CP-1901007 and date of approval: 2 January 2019). The anti-*SOX11* antibody (1:500 dilution, #ab229185) was purchased from Abcam (Abcam, UK). The tissue microarray (HOrgC120PG04) contains 120 paraffin-embedded tissue samples. Among them, 21 samples were either invalid or missing. A total of 99 valid samples were ultimately obtained, which included 45 malignant tumor tissues and 54 adjacent tissues of cancer or normal tissues. We calculated the *SOX11* immunostaining score by multiplying the percentage of positively stained cells with the intensity of staining. In this study, the high expression of *SOX11* was explained by a staining index score of ≤6, while the low expression of SOX11 was defined as a staining index of >6. The protein level of *SOX11* in cancer and corresponding normal tissues was also validated in Human Protein Atlas (HPA) database (https://www.proteinatlas.org/, accessed on 4 September 2022).

### 2.3. Relationship of SOX11 Expression with Survival Outcomes and Clinical Characteristics

We acquired the survival and clinical data of pan-cancer from the UCSC XENA repository. Kaplan–Meier method and Cox proportional hazards model were adopted to assess the correlations between *SOX11* expression and OS, progression-free survival (PFS), disease-specific survival (DSS) and disease-free survival (DFS). Kaplan–Meier survival curves were generated by the “survival” and “survminer” packages. Here, we also explored the associations of *SOX11* expression with tumor stage and tumor status.

### 2.4. Genomic Alterations of SOX11 in Pan-Cancer and Breast Cancer

*SOX11* gene alterations in TCGA pan-cancer datasets were examined by the cBioPortal database (http://www.cbioportal.org/, accessed on 10 September 2022) [29]. The “Oncoprint” and “Cancer Type Summary” tabs were used to explore the genetic alterations of *SOX11*. The mutated site information of *SOX11* was analyzed by the “Mutations” module in the database. The effect of *SOX11* alterations on disease-free survival and disease-specific survival was investigated using the “Comparison/Survival” module.

### 2.5. Analysis of SOX11 Expression with Stemness Score and EMT-Related Genes in Cancers

The RNA stemness score (RNAss) or DNA stemness score (DNAss) based on mRNA expression could be used to assess tumor stemness [30]. We intersected *SOX11* gene expression data with RNAss or DNAss to calculate the correlation coefficient. The results were visualized with the corrplot R package. In addition, correlation analysis between *SOX11* expression and cancer stem cell marker genes and EMT-related genes was further determined. A heatmap graph was created using the R packages “reshape2” and “RColorBrewer.”

### 2.6. Analysis of SOX11 Expression with TMB and MSI in Pan-Cancer

Biomarkers such as TMB and MSI are considered predictive of immune checkpoint inhibitor sensitivity [31,32]. We first calculated the TMB scores of each sample using TCGA pan-cancer mutation data. The MSI data in this study were obtained from Bonneville et al.’s pan-cancer study [33]. The correlations between *SOX11* expression and TMB, MSI were tested using Spearman’s coefficient. The results were presented in radar plots by the R-package “fmsb.” A defect in the DNA mismatch repair system (MMR) was defined as the presence of microsatellite instability (MSI-H) and/or the lack of hMLH1 protein expression [34]. We thus explored the relationship between *SOX11* expression and MMR-related genes in cancers. The heatmaps were generated using the R packages “RColorBrewer” and “reshape2”.

### 2.7. Relationship of SOX11 Expression with the Tumor Immune Microenvironment in Pan-Cancer

In this study, the StromalScore and ImmuneScore were calculated using the Estimation of Stromal and Immune Cells in Malignant Tumors using the expression data (ESTIMATE) algorithm to evaluate the level of estimated stromal and immune cells [35]. The corrplot R package was used to visualize Spearman rank correlation tests. Immune cell infiltration analysis was conducted using the CIBERSORT algorithm [36].

### 2.8. Gene Set Enrichment Analysis (GSEA)

To explore the potential biological functions of *SOX11* cancer, GSEA was performed using the R-packages “enrichplot”, “org.Hs.eg.db”, “clusterProfiler”, “DOSE”, and “limma”. The gene signatures of Gene ontology (GO) and Kyoto Encyclopedia of Genes and Genomes (KEGG) were compiled from the MSigDB database (https://www.gsea-msigdb.org/gsea/downloads.jsp, accessed on 12 August 2022).

### 2.9. Statistical Analysis

In this study, R statistical software version 4.1.1 (Vienna, Austria) was used to perform statistical analyses. The Cox proportional hazard regression model was used to estimate the hazard ratio (HR). Survival curves were compared using the Kaplan-Meier method and a log-rank test. Spearman rank correlation was used for the correlation analysis. Fisher’s exact test was applied to compare *SOX11* expression between cancer and normal tissues. The statistical significance of the results was defined as a *p*-value less than 0.05 (* *p* < 0.05, ** *p* < 0.01, *** *p* < 0.001).

## 3. Results

### 3.1. Expression Levels of SOX11 in Normal Tissues and Pan-Cancer

We first investigated the expression of *SOX11* in the normal physiological state of the body using the GTEX dataset. The results showed that *SOX11* was highly expressed in the brain, colon, heart, pituitary and spleen tissues, while a low or undetectable level of *SOX11* expression was found in other human tissues (Figure 1A). In Figure 1B, we explored the difference in *SOX11* expression between male and female normal tissues, and a significant sex difference in *SOX11* expression was found only in breast tissues. Then RNA-seq data from the TCGA database were applied to analyze the expression of *SOX11* in various cancers. Aberrant expression of *SOX11* was detected in a total of 27 cancers. *SOX11* was upregulated in adrenocortical carcinoma (ACC), bladder urothelial carcinoma (BLCA), breast invasive carcinoma (BRCA), cervical squamous cell carcinoma and endocervical adenocarcinoma (CESC), cholangiocarcinoma (CHOL), lymphoid neoplasm diffuse large B-cell lymphoma (DLBC), esophageal carcinoma (ESCA), glioblastoma multiforme (GBM), head and neck squamous cell carcinoma (HNSC), kidney chromophobe (KICH), kidney renal clear cell carcinoma (KIRC), brain lower grade glioma (LGG), liver hepatocellular carcinoma (LIHC), lung adenocarcinoma (LUAD), lung squamous cell carcinoma (LUSC), ovarian serous cystadenocarcinoma (OV), pancreatic adenocarcinoma (PAAD), prostate adenocarcinoma (PRAD), skin cutaneous melanoma (SKCM), stomach adenocarcinoma (STAD), testicular germ cell tumors (TGCT), thyroid carcinoma (THCA), thymoma (THYM), uterine corpus endometrial carcinoma (UCEC) and uterine carcinosarcoma (UCS) compared to normal tissues. In contrast, *SOX11* expression was significantly downregulated in colon adenocarcinoma (COAD) and rectum adenocarcinoma (READ; Figure 1C).

### 3.2. Protein Expression of the SOX11 in Human Tissues

We then detected the protein level of *SOX11* using human pan-cancer tissue arrays. According to the results, *SOX11* was expressed in the normal stomach (*n* = 4; Figure 2C), pancreas (*n* = 4; Figure 2G) and kidney (*n* = 3; Figure 2K) tissues, and it was slightly expressed in the normal colon tissues (*n* = 4; Figure 2D). *SOX11* expression was also found in the cancer tissues of the thyroid (*n* = 2; Figure 2A), esophagus (*n* = 3; Figure 2B), colon (*n* = 6; Figure 2D), rectum (*n* = 4; Figure 2E), liver (*n* = 5; Figure 2F), lung (*n* = 7; Figure 2H,I) and breast (*n* = 4; Figure 2J) tissues. The expression of *SOX11* was higher in ESCA, LUSC and LUAD than in normal tissues. In contrast, *SOX11* was more lowly expressed in KIRC than in the corresponding normal tissues. *SOX11* tended to have a higher expression in COAD and BRCA, while a lower expression of *SOX11* was also found in STAD. But the difference was not statistically significant due to the small sample size. The semiquantitative immunohistochemical results and statistical analysis were summarized in Appendix A. The result was not consistent with the mRNA expression level of COAD, STAD, PAAD and KIRC of TCGA data. We speculated that it might be due to the small sample size or tumor heterogeneity. We also verified the *SOX11* protein level in cancer and corresponding normal tissues using the HPA database (Appendix A).

### 3.3. Prognostic Value of SOX11 in Pan-Cancer

To further explore the prognostic value of *SOX11* in pan-cancer, Kaplan-Meier analysis was used to assess the relationship between *SOX11* expression level and different survival outcomes. Kaplan-Meier analysis showed that high expression of *SOX11* is a risk factor for survival in certain tumors. First, high *SOX11* expression was correlated with ACC (*p* < 0.001, Figure 3A), kidney renal papillary cell carcinoma (KIRP; *p* < 0.001, Figure 3B), LIHC (*p* < 0.001, Figure 3C), pheochromocytoma and paraganglioma (PCPG; *p* = 0.036, Figure 3D), sarcoma (SARC; *p* = 0.024, Figure 3E) and UCEC (*p* = 0.005; Figure 3F) in terms of overall survival. Regarding PFS, higher *SOX11* expression levels in ACC (*p* < 0.001; Figure 3G), BRCA (*p* = 0.021; Figure 3H), KIRC (*p* = 0.016; Figure 3I), LIHC (*p* < 0.001; Figure 3J), PRAD (*p* < 0.001; Figure 3K), SARC (*p* < 0.05; Figure 3L) and UCEC (*p* < 0.05, Figure 3M) were associated with shorter PFS. In addition, patients with high levels of *SOX11* expression also had significantly shortened DFS in ACC (*p* = 0.013; Figure 4A), KIRP (*p* = 0.028; Figure 4B), LIHC (*p* = 0.008; Figure 4C), PAAD (*p* = 0.009; Figure 4D) and PRAD (*p* < 0.001; Figure 4E). In addition, the same results were observed for DSS, especially in ACC (*p* < 0.001; Figure 4F), BRCA (*p* = 0.003; Figure 4G), KIRC (*p* = 0.006; Figure 4H), KIRP (*p* = 0.005; Figure 4I), LIHC (*p* < 0.001; Figure 4J) and UCEC (*p* < 0.001; Figure 4K).

### 3.4. Associations between the SOX11 Expression with Clinicopathological Features in Pan-Cancer

We compared the differences in *SOX11* expression levels at different tumor stages. In ACC (Figure 5A), COAD (Figure 5B), ESCA (Figure 5C), TGCT (Figure 5D) and UVM (Figure 5E), the expression of *SOX11* showed an increasing trend with increasing tumor stage. It has also been known that post-treatment tumor status was strongly associated with disease recurrence. We found that high expression of *SOX11* was significantly associated with tumor status in ACC (Figure 5F), PRAD (Figure 5G), UCEC (Figure 5H) and UVM (Figure 5I).

### 3.5. Analysis of Genetic Alterations in the SOX11 Gene of Pan-Cancer Patients

Next, we further investigated the genomic alterations of *SOX11* using the TCGA pan-cancer dataset. As shown in Figure 6A, the *SOX11* gene was found to be altered in only 1.4% of the TCGA pan-cancer dataset. As for the types of genetic alterations in *SOX11* in different cancers, mutations were the most frequent, particularly in UCEC, LUSC, CESC, STAD, ESCA, COAD/READ, mesothelioma (MESO), ACC, PAAD and HNSC. Gene amplifications ranked second, and they occurred in patients with UCS, LIHC, OV, BLCA, PRAD, GBM and BRCA. Deep deletions were the most common type of genetic alteration in patients with DLBC and KICH (Figure 6B). The types, numbers and sites of the *SOX11* genetic alterations are displayed in Figure 6C. Notably, missense mutations were the most common type of mutation, and it was obvious that the mutated site was located in the HMG structural domain of *SOX11*. Next, genetic alterations in *SOX11* were analyzed in TCGA-BRCA, where the *SOX11* gene was altered in 9% of all breast cancer samples. The highest frequency of genetic alterations was mRNA high expression (Figure 6D). Regarding the impact of *SOX11* gene alterations on survival in BRCA patients, we found that patients with the *SOX11* gene-altered group had significantly shorter DSS and PFS than those with the *SOX11* gene unaltered group (Figure 6E,F).

### 3.6. Relationship between the SOX11 Expression and EMT-Related Genes and Stemness Scores in Pan-Cancer

Next, we investigated the correlation between *SOX11* expression and stemness. First, we explored the association between *SOX11* expression levels and stem cell scores in pan-cancer. Correlation analysis showed that *SOX11* expression levels were positively correlated with DNAss in GBM, SARC and THCA but negatively correlated with DNAss in OV, PCPG and TGCT (Figure 7A). Furthermore, *SOX11* expression levels were positively correlated with RNAss in BRCA, GBM and LGG but negatively correlated with those in COAD, HNSC, KIRC, PCPG and READ (Figure 7B). In addition, the heatmap showed positive correlations between the expression of more than 15 EMT-related genes and the expression of *SOX11* in BLCA, BRCA, COAD, HNSC, KICH, KIRC, LGG, LIHC, LUSC, PAAD, READ, SARC, SKCM and TGCT (Figure 7C). Figure 7D describes the correlation between *SOX11* expression and 18 tumor stem cell markers. In BRCA, COAD, GBM, LGG, LIHC, LUSC, OV, SARC, SKCM and TGCT, there was a significant positive correlation between more than 10 tumor stem cell markers and the expression of *SOX11*.

### 3.7. Correlation between SOX11 Expression and MSI, TMB and MMR-Related Genes in Cancer

The expression levels of *SOX11* were further investigated for correlation with TMB, MSI and MMR in pan-cancer. The results showed that *SOX11* expression was significantly correlated with increased TMB in BRCA, BLCA, ACC, TGCT, PRAD, MESO and LGG and with decreased TMB in UCS, UCEC, LUAD, KIRP and HNSC (Figure 8A). Regarding MSI, the level of *SOX11* was positively correlated with increased MSI in BRCA, TGCT, SARC, PRAD and OV, while they were negatively correlated in SKCM and KIRC (Figure 8B). As shown in Figure 8C, *SOX11* expression was significantly positively correlated with the expression of MMR-related genes in most tumors, particularly in BRCA, GBM, LGG, LIHC, LUAD and OV.

### 3.8. Relevance of SOX11 Expression to the Tumor Immune Microenvironment

Currently, the predictive role of *SOX11* in the tumor immune microenvironment remains unknown. Here, the correlation between *SOX11* expression and the tumor immune microenvironment was assessed using the ESTIMATE method. Our results showed that *SOX11* expression was positively correlated with the ESTIMATE scores in BLCA, COAD, PCPG, READ, THCA and UVM, as well as immune scores (Figure 9A,B). We also explored the relationship between *SOX11* expression and immune cell infiltration in pan-cancer. The results showed that *SOX11* expression correlated with the level of infiltration of all 15 immune cell types, including B cells, plasma cells, CD4+ T cells, CD8+ T cells, M0 macrophages, M1 macrophages, M2 macrophages, monocytes, NK cells, neutrophils, follicular helper T cells, γΔT cells, Treg cells, dendritic cells, and mast cells (Appendix A). We found that in COAD, HNSC, LUAD, PAAD, SKCM, THCA and THYM, the infiltration levels of CD8+ T cells were all significantly negatively correlated with the expression levels of *SOX11* (Figure 9C–I).

### 3.9. The Biological Function of SOX11 in Pan-Cancer

The main biological processes of *SOX11* in cancers were investigated by GSEA. Regarding the KEGG pathways analysis, our results demonstrated that *SOX11* was positively correlated with the metabolism-related pathways in OV, such as the ascorbate and aldehyde metabolism, pentose and glucuronate interconversion and drug metabolism cytochrome P450. However, *SOX11* was negatively enriched in metabolism-related pathways of KIRC, such as the citrate cycle, folate biosynthesis and steroid biosynthesis. Moreover, *SOX11* positively regulated immune-related pathways in PAAD, PRAD and STAD, such as antigen presentation and the RIG-I-like receptor signaling pathway. *SOX11* was also found to be positively associated with disease-related pathways, including dilated cardiomyopathy, hypertrophic cardiomyopathy and maturity-onset diabetes of the young. In addition, *SOX11* was identified as a positive regulator of olfactory transduction and autophagy in HNSC, KIRC, LUSC, OV, PAAD, PRAD and STAD. We also found that *SOX11* is closely associated with receptor-related pathways, including negative regulation of the Nod-like and Toll-like receptor signaling pathway in GBM, but positive regulation of these receptor-related pathways in LUSC, PAAD, PRAD and STAD. (Figure 10) The GO results of GSEA analysis of *SOX11* in pan-cancer were shown in Appendix A. *SOX11* was positively enriched in ACC, COAD, DLBC, ESCA, GBM, acute myeloid leukemia (LAML), LGG, LUAD, LUSC, MESO, OV, PCPG, PRAD, STAD, THYM and UCEC in gene regulatory mechanisms such as gene silencing, mRNA-mediated gene silencing, miRNA-mediated translational repression, mRNA binding and chromosome segregation regulation. In CHOL, KICH, KIRC and UVM, *SOX11* was also positively enriched in a variety of immune response pathways, such as regulation of cell surface receptor signaling, regulation of immune effector processes and production of molecular mediators of immune responses. In contrast, *SOX11* was negatively enriched in BLCA, CESC, KIRC, LIHC, LUAD, LUSC, PRAD, SKCM, TGCT and UCS in the regulation of cell cycle G1/S transition, epidermal cell growth, epidermal cell differentiation and epithelial cell migration.

## 4. Discussion

In this study, we found that *SOX11* was highly expressed in the brain, colon, heart, pituitary gland and spleen tissues, while it was expressed at lower levels in other normal human tissues. The results of tissue microarray showed that *SOX11* was positively expressed in the normal thyroid, esophagus, stomach, colon, rectum, liver, pancreas and kidney. Previous studies demonstrated that *SOX11* was highly expressed in the central nervous system, gastrointestinal tract, lung, spleen, pancreas, kidney and gonads in normal humans [12,13,14]. *SOX11* was reported to be highly expressed in breast cancer [17], mantle cell lymphoma [18], epithelial ovarian cancer [19] and hepatocellular carcinoma [20], while it was expressed at low levels in nasopharyngeal carcinoma [21], prostate cancer [22], gastric cancer (GC) [23] and bladder cancer [24]. However, we discovered that *SOX11* expression was altered in 27 cancer tissues from the TCGA database compared to normal tissues. The mRNA level of *SOX11* was upregulated in most of the cancer tissues except COAD and READ. In the tissue microarray, *SOX11* was more highly expressed in ESCA, LUSC and LUAD than in the corresponding normal tissues, but the results were reversed in KIRC. We also observed that *SOX11* tended to have a higher expression in COAD, READ, LIHC and BRCA than in the corresponding normal tissues, and the effect was reversed in STAD and PAAD. However, these differences did not reach statistical significance. Additionally, we found that *SOX11* was differentially expressed in male and female breast tissues, which might be due to the differences volume and structure of male and female mammary glands. In summary, compared to corresponding normal tissues, *SOX11* protein level was upregulated in ESCA, THCA, COAD, READ, LIHC, LUSC, LUAD and BRCA. In contrast, *SOX11* protein level was downregulated in STAD, PAAD and KIRC.

Regarding the prognostic value of *SOX11* in cancer, the available findings are controversial. *SOX11* was reported to act as both an oncogenic and a tumor suppressor. For example, *SOX11* was considered to be a key oncogenic factor in MCL [37]. And the high expression of *SOX11* was strongly associated with poor prognostic factors of chronic lymphocytic leukemia (CLL) [38]. Regarding the mRNA level of *SOX11* in DLBCL, a previous study also exhibited a low transcriptional level of *SOX11* in patients with DLBCL, which was consistent with our findings [39]. In breast cancer, high *SOX11* expression levels were also significantly associated with distant metastasis and poor prognosis [40]. Additionally, *SOX11* could promote brain metastasis in triple-negative breast cancer (TNBC) [41]. In hepatic cell carcinoma (HCC), higher mRNA transcript levels of *SOX11* were associated with shorter OS and PFS [42]. However, it was reported that the upregulation of *SOX11* expression might be a good prognostic factor for patients with MCL [18]. Additionally, *SOX11* was found to be an independent prognostic factor for improving survival in gastric cancer [43]. Studies revealed that the overexpression of *SOX11* inhibited the invasion of COAD [44], and the upregulation of *SOX11* also hindered the invasion and proliferation of LUAD cells [45]. In particular, high nuclear expression of *SOX11* was found to be associated with longer OS in breast cancer patients, which meant that it was an independent predictor of survival [46]. In this study, our Kaplan–Meier survival analysis showed that high expression of *SOX11* was significantly correlated with poor survival in patients with ACC, KIRP, LIHC, PCPG, SARC, UCEC, KIRC, PRAD, PAAD and BRCA, including OS, PFS, DFS and DSS. Our study suggested that *SOX11* was associated with poor survival in both HCC and BRCA, which contradicted some previous studies.

We next explored the relationship of *SOX11* expression with the clinicopathological parameters of patients. It was previously reported that high *SOX11* expression was positively correlated with breast cancer tumor size and early tumor grade and negatively correlated with lymph node metastasis in breast cancer [46]. Clinical studies also showed that *SOX11*-positive MCL patients with lymph node involvement had a poorer prognosis [47]. In addition, high mRNA levels of *SOX11* in HCC were closely associated with the high grade of the tumor [42]. In our study, *SOX11* expression levels in ACC, COAD, ESCA, TGCT and UVM showed an increasing trend with increasing tumor stage, but the trend was more pronounced mainly in tumor stages I-III which might be due to the small sample size of stage IV patients. In addition, we found that high expression of *SOX11* was significantly associated with tumor status in ACC, PRAD, UCEC and UVM, which indicated that *SOX11* might have the potential to reflect tumor progression.

Regarding genetic alterations of *SOX11* in pan-cancer, our data found that mutations were the most common type of genetic alteration of *SOX11*, followed by amplifications. *SOX11* mutations were reported to occur frequently in the hotspot(D233 del/2_D233 del) in esophageal, gastric and lung cancers, while amplification was the major form of genetic alteration in neuroendocrine prostate cancer (NEPC), OV, PRAD, BRCA and UCEC [48]. A study explored the genetic alterations of the *SOX* family in patients with HCC and found that all genes in the *SOX* family had some genetic alterations, including missense mutations, truncating mutations, amplifications, deep deletions and high mRNA [42]. It was important to note that *SOX11* amplification or mutation was uncommon in primary cancers [49], but *SOX11* amplification and upregulation of expression were found in metastatic brain cancers [50], which was consistent with the results of our study. We found that *SOX11* mutations were most seen in esophageal, gastric and lung cancers, while *SOX11* amplification was most commonly seen in liver cancer. However, the expression level and genetic alterations of *SOX11* in metastatic cancers remain unknown. In addition, our study found for the first time that *SOX11* amplification was the most common form of gene alteration in BLCA and GBM, and the deep deletion of *SOX11* gene was most common in DLBC and KICH. Our study also identified missense mutations located in the high-mobility-group (HMG) domain of *SOX11*, which was also consistent with previous reports [51]. What’s more, patients in the *SOX11* altered group had shorter DSS and PFS than patients in the *SOX11* unaltered group, suggesting that *SOX11* gene alterations, particularly high mRNA levels, were risk factors for prognosis in breast cancer. However, the mechanisms were not clear, which might need to be further explored in future experimental and clinical studies.

Cancer stem cells (CSCs), a subpopulation of cancer cells, have similar characteristics to normal stem/progenitor cells, such as the ability to self-renew and differentiate to drive tumor growth [52]. Our study found that *SOX11* expression was highly correlated with stem cell markers as well as EMT-related genes. A previous study confirmed that *SOX11* was identified as a characteristic gene of mesenchymal stem cells (MSCs) and a potential biomarker for early progenitor human MSCs [53]. Microarray analysis showed that *SOX11* mRNA gradually decreased during the expansion of bone marrow MSCs, suggesting that *SOX11* could be used as a biomarker to distinguish early progenitor cells from mature bone marrow MSCs [54]. It was found that *SOX11* was a better biomarker of breast cancer stem cells that could predict cancer recurrence. Furthermore, *SOX11* expression was directly associated with breast cancer stem cell populations and correlated with overexpression of *ALDH1* [48]. Bhattaram et al. suggested that SOXC proteins might control the fate and behavior of stem/progenitor cells in many lineages through a mechanism distinct from other SOX proteins [55]. EMT is a common process in tumorigenesis and has long been considered to be an important embryonic process. *SOX11* was upregulated in TGF-β and EGF-induced EMT in renal tubular epithelial cells, suggesting a possible involvement of *SOX11* in EMT [56]. Another study showed that *SOX11* could activate *SLUG* expression in endocrine-resistant breast cancer cell lines by binding to its promoter, thereby promoting EMT [57].

Evidence suggests that MSI and TMB could serve as potential indicators of response to immunotherapy [58]. Notably, immunotherapies have also shown efficacy for CRC patients with mismatch-repair-deficient and microsatellite instability-high (dMMR-MSI-H) [59]. It has been reported that *SOX11* hypermethylation is closely associated with MSI in endometrial cancer [60]. Our results showed that *SOX11* expression positively correlated with elevated MSI in BRCA, TGCT, SARC, PRAD and OV, as well as with elevated TMB in BRCA, BLCA, ACC, TGCT, PRAD, MESO and LGG. The results above indicated that *SOX11* might also serve as a novel and effective biomarker for predicting the efficacy of immunotherapy. The TME is defined as the cellular environment in which a tumor is located, grows and expands. Tumor cells release extracellular factors such as vascular endothelial growth factor, molecules that promote tumor angiogenesis or induce peripheral immune tolerance [61]. Balsas et al. showed that *SOX11* regulated MCL homing and invasion by directly regulating *CXCR4* and *FAK* expression [62]. The correlation between *SOX11* expression and immune cell content might depend on tumor cell type according to the ESTIMATE algorithm. Studies found that intra-tumoral infiltrations of T cells were significantly reduced in *SOX11+* MCL [63]. Further analysis revealed that *SOX11* expression correlated significantly with the level of infiltration of various immune cells, particularly in BRCA, CESC, COAD, HNSC and KIRC. In COAD, HNSC, LUAD, PAAD, SKCM, THCA and THYM, CD8+ T cell infiltration levels were all significantly negatively correlated with *SOX11* expression levels, suggesting that *SOX11* might be associated with immune escape. At present, less research has been done regarding the role of *SOX11* in immune regulation. The relationship between *SOX11* expression and immune infiltration requires further investigation in vitro and in vivo.

Our GSEA analysis showed that *SOX11* expression was closely associated with metabolism-related pathways, immune-related pathways, disease-related pathways and receptor-related pathways. GO analysis showed that *SOX11* expression was positively enriched in gene regulatory mechanisms as well as multiple immune response pathways, while *SOX11* expression was negatively enriched in the regulation of cell cycle G1/S transition, epidermal cell growth, epidermal cell differentiation and epithelial migration. *SOX11* has been reported to affect apoptosis-associated cell survival pathways, *TEAD2* transcriptional activity, mitosis and immune-inflammatory responses [15,48,64]. It has also been shown that *SOX11* may be associated with specific key pathways related to certain developmental processes, including the WNT signaling pathway and the TCF/β-linked protein complex in hepatocellular carcinoma [65]. In addition, we found that *SOX11* could activate the bone morphogenetic protein (BMP)/Smad signaling pathway in mesenchymal stem cells (MSCs) and could transcriptionally activate runt-related transcription factor 2 (*Runx2*) and CXC chemokine receptor-4 (*CXCR4*) expression [53]. Our study also identified that one signaling pathway could be significantly enriched in multiple tumor types, such as gene silencing, miRNA-mediated translational repression, mRNA binding and chromosome segregation regulation. Thus, we suggested that *SOX11*-mediated pathways in different cancers were not specific.

The significance of this study is to reveal the prognostic value of *SOX11* in pan-cancer, which not only further supports previous findings but also contributes to our understanding of the multifaceted role of *SOX11* in tumor progression. As our study conducted a comprehensive bioinformatics analysis that relies on multiple databases, there were inevitably some limitations of our analyses. First, further experimental validation is required in future studies to confirm these results. Second, most of our analyses focused on *SOX11* mRNA expression. It is worth mentioning that analysis based on *SOX11* protein levels would enhance the credibility of the results. Third, only relevant analyses were performed in this study, and the molecular mechanisms of *SOX11* in tumor stemness and immune infiltration require further investigation.

## 5. Conclusions

In summary, this study systematically explored the characteristics of *SOX11* in terms of expression patterns, survival prognosis, gene mutations, stemness, TMB, MSI, immune infiltration and related signaling pathways. *SOX11* may be a potential biomarker in cancer, as it exhibits aberrantly high expression in a variety of cancers and predicts poor prognosis in cancer patients. In different cancers, a large number of mutations, amplifications and high mRNA levels were observed in *SOX11*. In addition, aberrant *SOX11* expression was associated with stem cell scoring, EMT-related genes, MSI, TMB and the tumor immune microenvironment in different cancer types. Finally, our work also suggests several future directions for future prospective studies that focus on *SOX11* in cancer.

## Figures and Tables

**Figure 1 cancers-14-06103-f001:**
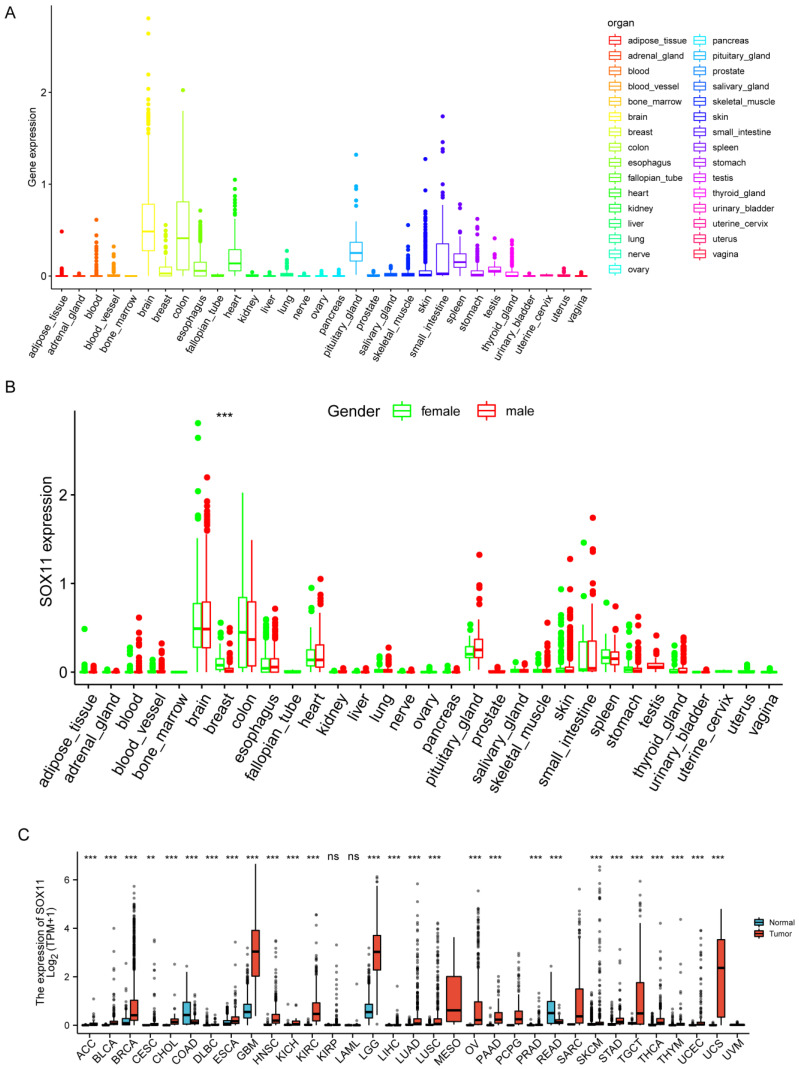
Expression analysis of *SOX11*. (**A**) The *SOX11* expression in normal tissues from Genotype Tissue-Expression (GTEX) data. (**B**) The *SOX11* expression abundances of various tissues in males and females. ***, *p* < 0.001. (**C**) Differential *SOX11* mRNA expression between 33 TCGA cancers and GTEX normal tissues. The red color column represents the cancer samples, and the blue color column represents the normal samples. ns, *p* ≥ 0.05; **, *p* < 0.01; and ***, *p* < 0.001.

**Figure 2 cancers-14-06103-f002:**
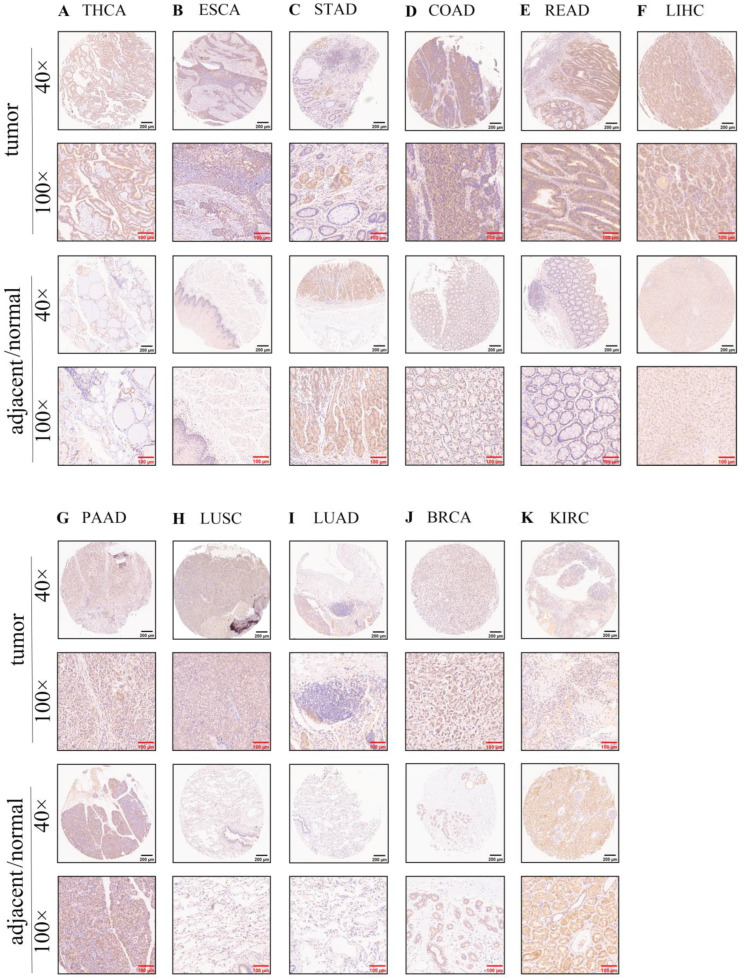
Protein expression level of *SOX11* in human multiple cancer tissues and normal tissues of THCA (**A**), ESCA (**B**), STAD (**C**), COAD (**D**), READ (**E**), LIHC (**F**), PAAD (**G**), LUSC (**H**), LUAD (**I**), BRCA (**J**) and KIRC (**K**). Representative images of *SOX11* expression in pan-cancer tissues are shown. Original magnification, ×40 and ×100, Scale bar: black, 200 μm; red, 100 μm.

**Figure 3 cancers-14-06103-f003:**
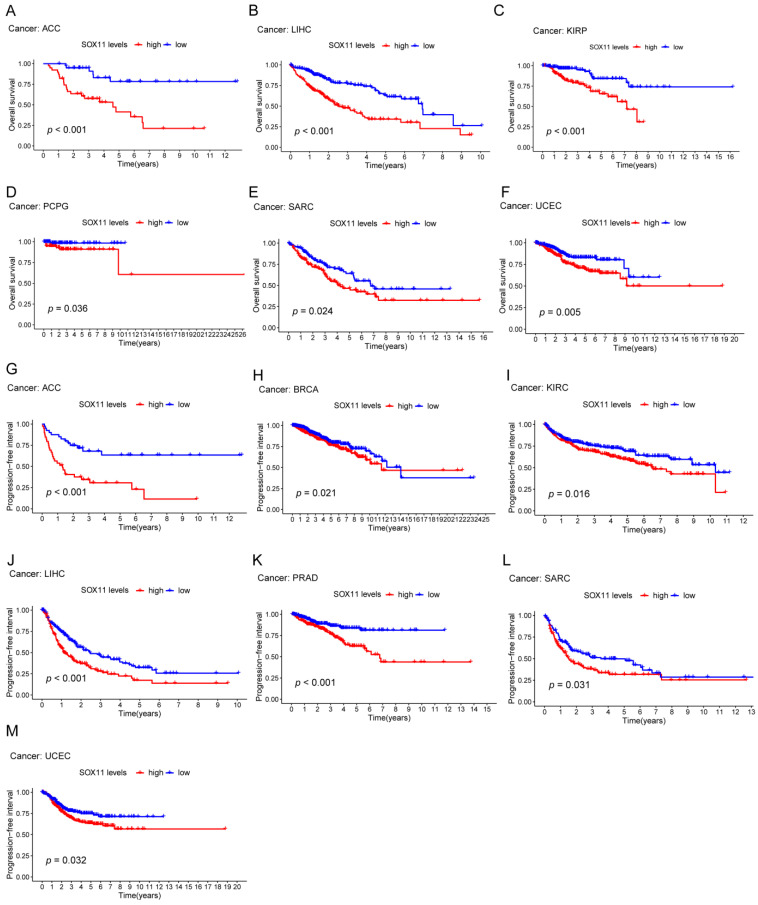
The Kaplan–Meier analysis of the correlations between the *SOX11* expression and OS and PFS. (**A**–**F**) The Kaplan–Meier analysis of the correlations between the *SOX11* expression and OS. (**G**–**M**) The Kaplan–Meier survival analysis shows the impact of *SOX11* expression on PFS. The red line indicates a high *SOX11* expression group, and the blue line indicates a low *SOX11* expression group.

**Figure 4 cancers-14-06103-f004:**
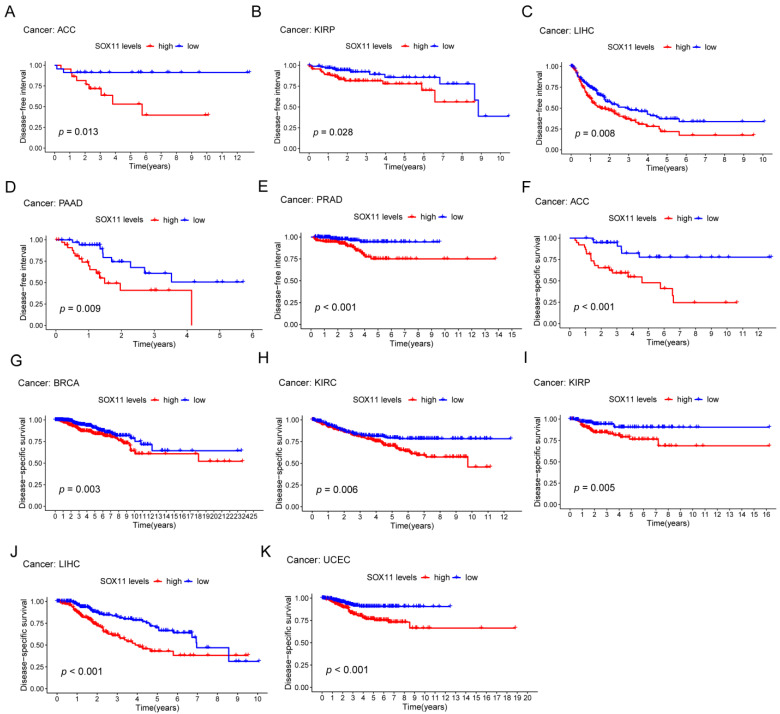
The Kaplan–Meier analysis of the correlations between the *SOX11* expression and DFS and DSS. (**A**–**E**) The Kaplan–Meier survival analysis shows the impact of *SOX11* expression on DFS. (**F**–**K**) The Kaplan–Meier survival analysis shows the impact of *SOX11* expression on DSS. The red line indicates a high *SOX11* expression group, and the blue line indicates a low *SOX11* expression group.

**Figure 5 cancers-14-06103-f005:**
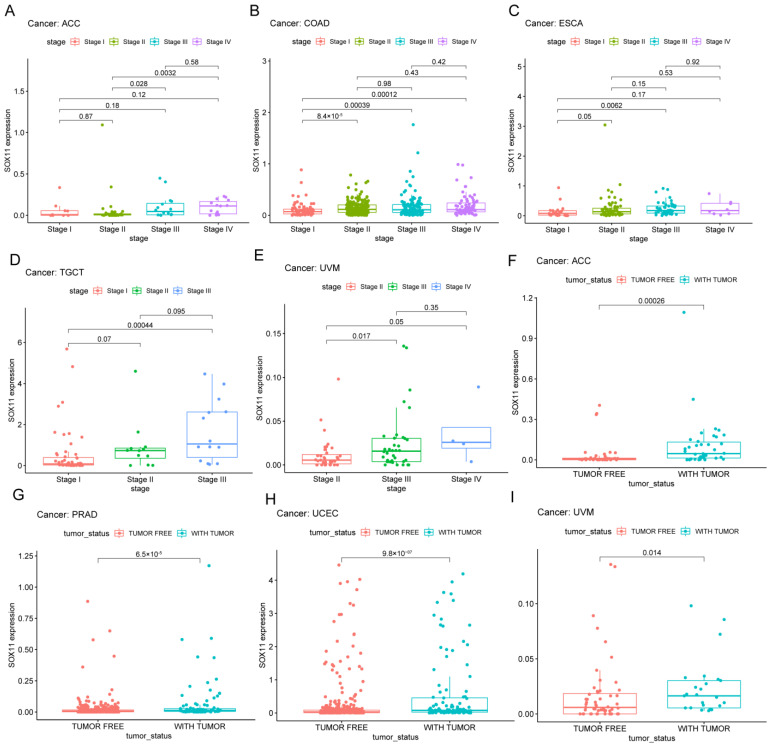
Correlation between *SOX11* expression and Clinicopathological Phenotype. (**A**–**E**) Correlation between *SOX11* expression and clinical stage. (**F**–**I**) Correlation between *SOX11* expression and tumor status.

**Figure 6 cancers-14-06103-f006:**
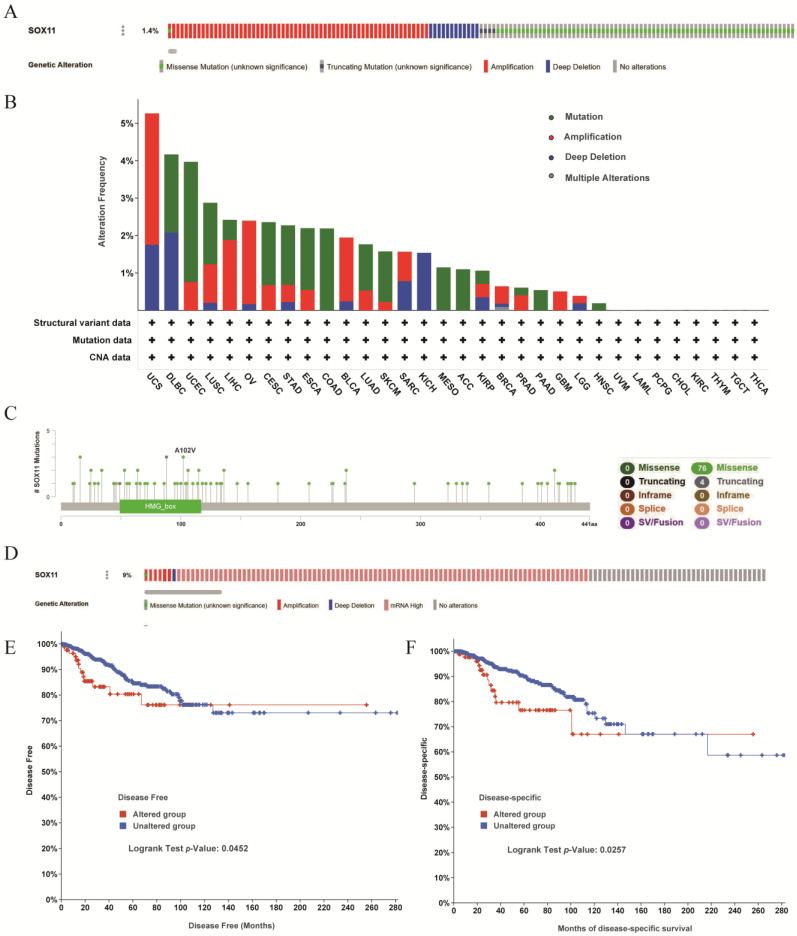
Pan-cancer *SOX11* alterations from The Cancer Genome Atlas (TCGA). (**A**) OncoPrint visual summary of *SOX11* alterations in TCGA pan-cancer datasets. Four types of genetic alterations were defined: Missense Mutation (unknown significance), Truncating Mutation (unknown significance), Amplification and Deep Deletion (**B**) A comprehensive view of the alteration frequency of *SOX11* in TCGA pan-cancer datasets. “+” represents that the dataset contained the indicated data. (**C**) Analysis of the mutation sites and mutation types in *SOX11*. (**D**) OncoPrint summary of the alterations on *SOX11* in TCGA BRCA dataset. Four types of genetic alterations were defined: Missense Mutation (unknown significance), Amplification, mRNA high and Deep Deletion (**E**) Kaplan–Meier analysis of disease-free survival in TCGA BRCA cohort with altered or unaltered *SOX11*. (**F**) Kaplan–Meier analysis of disease-specific survival in TCGA BRCA cohort with altered or unaltered *SOX11*.

**Figure 7 cancers-14-06103-f007:**
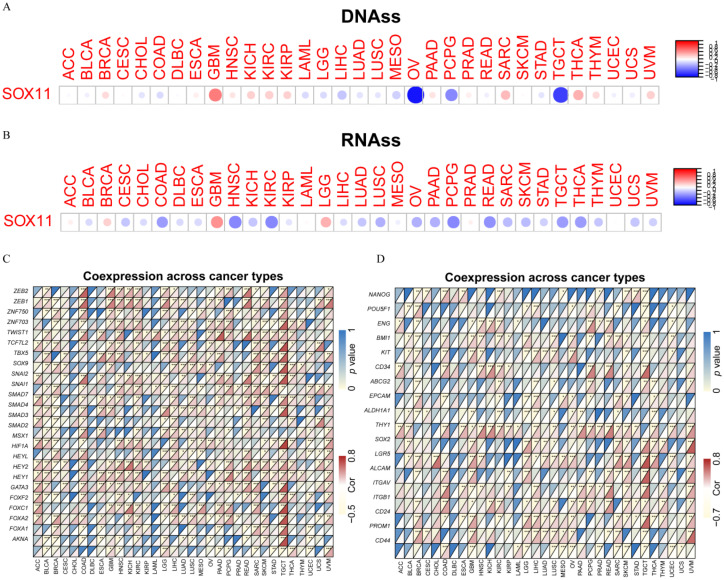
Analysis of the relationship between *SOX11* expression and stemness and EMT-related genes in cancers. (**A**) The correlation between *SOX11* expression and DNAss in different cancers. The red dots indicate positive correlations, while the blue dots denote negative correlations. DNAss, DNA stemness score (**B**) The correlation between *SOX11* expression and RNAss in different cancers. RNAss, RNA stemness score (**C**) Correlation heatmap showing the relationship between *SOX11* expression and EMT-related genes expression. For each pair, the top left triangle indicates the *p*-value, and the bottom right triangle denotes the correlation coefficient. (**D**) Correlation heatmap showing the relationship between *SOX11* expression and cancer stem cell markers * *p* < 0.05, ** *p* < 0.01, and *** *p* < 0.001. Cor, correlation coefficient.

**Figure 8 cancers-14-06103-f008:**
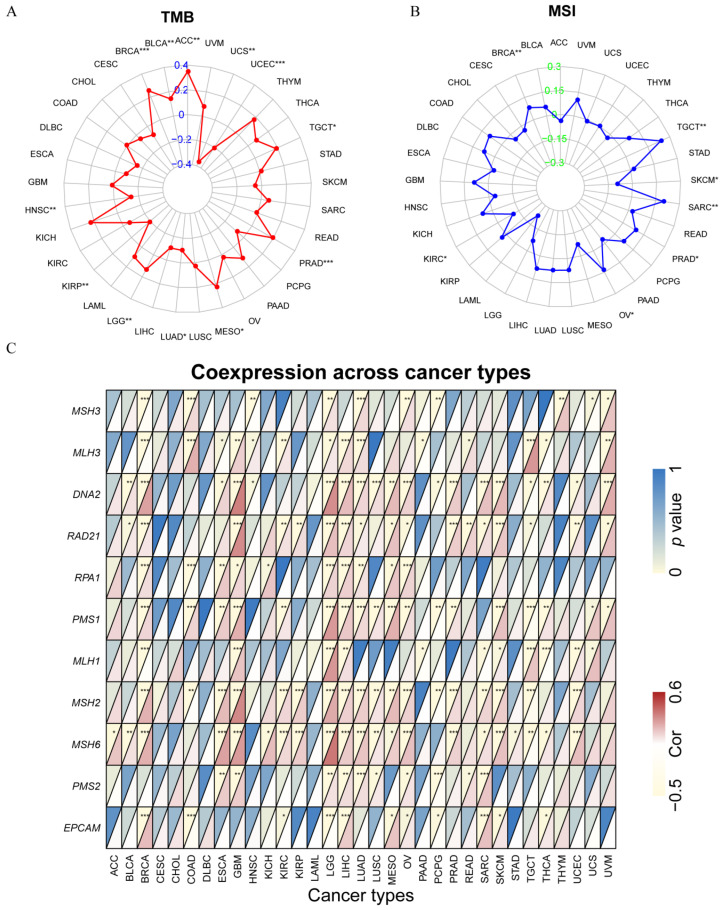
Analysis of the associations of *SOX11* expression with TMB, MSI and MMR genes. (**A**) Radar map reflecting the correlation between the *SOX11* expression and TMB. (**B**) Radar map reflecting the correlation between the *SOX11* expression and MSI. (**C**) Correlation heatmap showing the relationship between *SOX11* expression and MMR genes. For each pair, the top left triangle indicates the p-value, and the bottom right triangle denotes the correlation coefficient * *p* < 0.05, ** *p* < 0.01, and *** *p* < 0.001. Cor, correlation coefficient.

**Figure 9 cancers-14-06103-f009:**
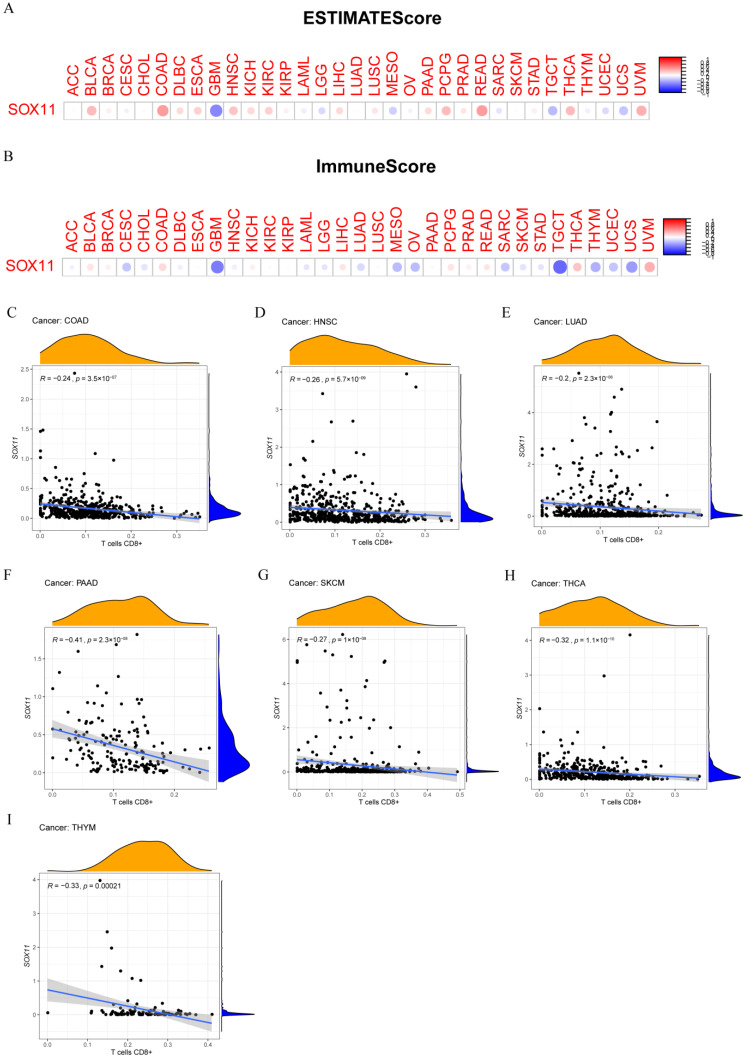
Analysis of the association of *SOX11* gene expression with the immune microenvironment. (**A**) An analysis of the relationship between *SOX11* expression and ESTIMATEScore in different cancers. The red dots indicate a positive correlation, while the blue dots denote indicate a negative correlation. (**B**) Correlation between the *SOX11* expression and ImmuneScore in different cancers. (**C**–**I**) Correlation between CD8+ T cell infiltration and *SOX11* expression in COAD, HNSC, LUAD, PAAD, SKCM, THCA and THYM.

**Figure 10 cancers-14-06103-f010:**
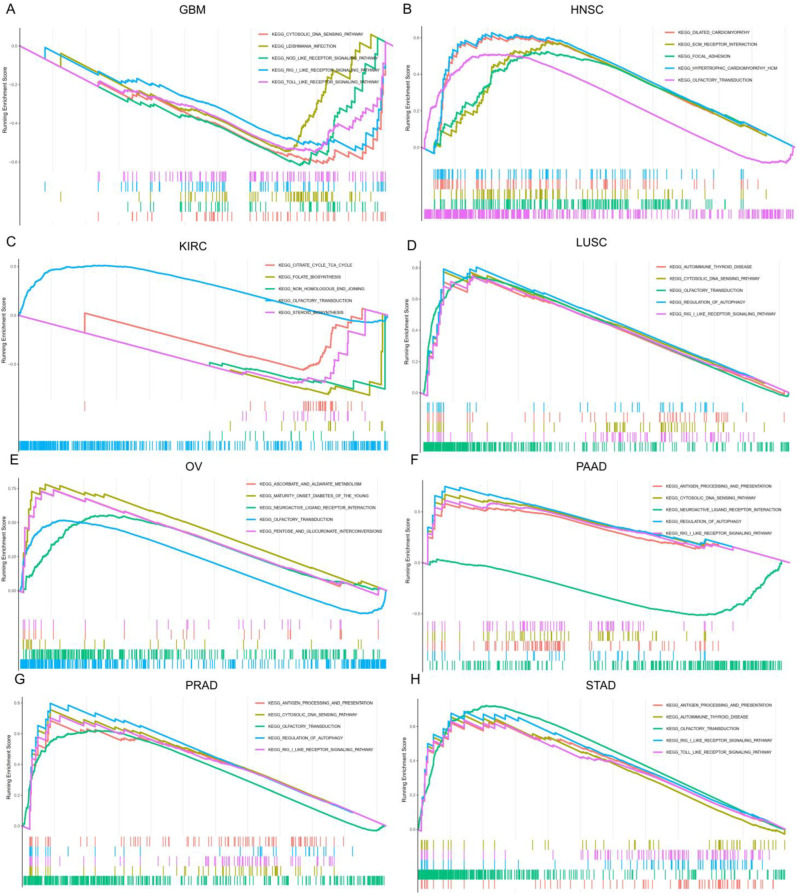
The Gene Set Enrichment Analysis (GSEA) in Kyoto Encyclopedia of Genes and Genomes (KEGG) signature of GBM, HNSC, KIRC, LUSC, OV, PAAD, PRAD and STAD. The curves with different colors indicate the different pathway annotations. Peaks in the upward direction indicate positive regulation by *SOX11* and peaks in the downward direction represent negative regulation by *SOX11*. (**A**) The GSEA analysis of *SOX11*-mediated biological process in GBM; (**B**) The GSEA analysis of *SOX11*-mediated biological process in HNSC; (**C**) The GSEA analysis of *SOX11*-mediated biological process in KIRC; (**D**) The GSEA analysis of *SOX11*-mediated biological process in LUSC; (**E**) The GSEA analysis of *SOX11*-mediated biological process in OV; (**F**) The GSEA analysis of *SOX11*-mediated biological process in PAAD; (**G**) The GSEA analysis of *SOX11*-mediated biological process in PRAD; (**H**) The GSEA analysis of *SOX11*-mediated biological process in STAD.

## Data Availability

All the data are available from the TCGA database and cBioPortal database.

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
