# Peer review of "Systematic Investigation of the Multifaceted Role of SOX11 in Cancer"

_cancers, 2022, doi:10.3390/cancers14246103_

Round 1

Reviewer 1 Report

The manuscript presented by Sun and coworkers is the first work englobing the study of SOX11 expression in all cancers with Gene expression data deposited at the TCGA Pan-cancer UCSC-XENA compared to the GE data from normal human tissue deposited at the GTEX website. The authors validated SOX11 expression by Immunohistochemistry in some of these cancers using tissue samples. The correlation of SOX11 expression with cancer grade, outcome (Overall Survival), and with stemness, microsatellite instability (MSI), tumor mutation burden (TMB), mismatch repair (MMR) related genes and tumor immune microenvironment has been analyzed in each cancer type. A global gene set enrichment analysis (GSEA) using the KEGG pathway gene set has been analyzed determining the close association of SOX11 with functional pathways important in cancer progression like enrichment of metabolism and immune response, among other, related pathways.

The manuscript is well written, concise and the bioinformatics methodology well engaged to demonstrate that SOX11 is an important factor in the process of tumorigenesis in several types of cancer.

However some important issues have to be addresses:

1.      It has been fully demonstrated that SOX11 is not expressed in DLBCL. However the authors state that SOX11 is expressed in this lymphoma based on their significant differences with normal tissues. First, they don’t specify with normal tissue they have used. The authors need to verify the correct normal tissue used to compare with each cancer. The authors should define a correct threshold, using the experimental demonstration from other authors, to determine the threshold and define the cancers that are expressing SOX11.

2.      The authors should demonstrate the Correlation of the SOX11 expression with some of the characteristic stemness and EMT, Estimates and Immune-related genes in different cancers, by gene or protein expression, like they did for CD8+Tcell makers.

3.      The References should be revised as several papers did not correspond with the bibliography referred at the reference section. For example: Reference 37 and 20.....

Reviewer 2 Report

This present article by Sun et al did an elaborate investigation of SOX11 in human cancer samples. However, the data presented here is mostly illegible. Therefore, I cannot accept this in the present form.  I am in principle supportive of accepting this work for publication only after the following correction has been made in the current form. Therefore, I encourage authors to correct the figures and re-submit. 

Major

1.Figure 1 panel A and B the graph should be legible. If they cannot present as a single graph they can split and present legibly. 

2. Figure 2. Quantification has to be done and the Scale bar should be displayed. 

3. Figure 6 panel A and C and E should be legible..

4. Figure 10 all panels should be legible. 

5. Supporting data 1 & 2 also lacks legibility and scale bar and quantification is missing in supporting data 3.

Reviewer 3 Report

Sun et al have set out to investigate correlations between SOX11 expression and some generalized characteristics of many tumor types. Unfortunately, this work suffers from a few key issues, mainly the statistics involved in assessing SOX11 are poorly defined and appear to not be stringent enough to properly account for large sample sizes. Dramatically small P-values are reported for many conditions that do not appear different at an observational level. Furthermore, there is no validation associated with this work, leaving any conclusions to be specious at best (as the authors themselves admit these observations are not confirmed on line 534). Finally, there is no discussion on the homology of the broader SOX family. Can the authors provide any evidence that these observations are not due to upregulation of other SOX family members, such as SOX2? Due to sequence homology, it is difficult to distinguish SOX family members by antibody staining. Many times, sequencing reads to the HMG domain can align to many of the SOX family members, but only by direct testing of gene-specific regions can expression be truly quantified. Overall, this report is only a correlative study and with the little explanation of the statistics used, and with the lack of controls against other SOX family members readers will have a hard time determining the significance of these results.

Major issues

Line 85 – “provides a theoretical basis for using Sox11 as a new therapeutic strategy” This is a vast overstatement considering that the authors have just described the role of SOX11 in normal development and these investigations are corelative at best. How would they suggest targeting SOX11 therapeutically?

Figure 2 – Can the authors provide a secondary-only control (perhaps as supplemental data) for these stains? Many of the stains are unimpressive and the degree that any staining is specific and not background is unclear. Indeed, can the authors claim in most of these that the tumor staining is higher than the adjacent tissues (For example please see PAAD)?

Statistics are unclear. Are these t-tests or z-tests?

How are stemness scores and EMT scores defined?

Wouldn’t the authors expect GO or KEGG terms related to stemness or developmental pathways, considering the published roles of SOX11?

Line 531 – No role of SOX11 has been elucidated from this work

Minor Issues

Line 101-104 – What constitutes a valid/invalid tissue?

Figure font sizes are too small to be readable (please see Figure 10 as an example)

Figure 1C – What is the statistical test used here? This does not seem appropriate as some cancers (DLBC for example) do not appear to be significantly upregulated yet show a significant P-value according to the test used.

Line 192 – Please define “non-dominant coloration”

Reviewer 4 Report

Minor comments:

1)      The manuscript is poorly written. Several sentences are confusing and sometimes misleading. There are places where the wording is somewhat awkward. I would suggest using an English Language Editing Service to ensure that your work is written in correct scientific English. Also, I would suggest more proof-reading and editing, ideally by a native speaker.

2)      Figures are of poor quality.

Round 2

Reviewer 2 Report

In this revised version, authors tried to improve the presented data. However, no quantification was performed in their tissue section (Fig 2) and the scale bar provided in Fig 2 was not legible. Likewise, their supplementary fig1 needs quantification and statistical tests must be performed with appropriate controls.

I am in principle not supportive of accepting this work for publication in its present form.

Reviewer 3 Report

I thank the authors for their revision and willingness to improve this manuscript. However, the authors are still confounding correlation with causation. They oftentimes speak of SOX11 as a "mediator" of a particular cellular function, however they have only described that SOX11 is correlated with such activity. At no time have they demonstrated that SOX11 has mediated any of the functions described in this manuscript. Indeed, SOX11 may simply be upregulated in more aggressive tumors as a passenger mutation, where tumor development is driven by other factors (perhaps other SOX family members). Furthermore, any discussion of SOX11 as a potential target for therapy is extremely premature without showing causation of any of the phenotypes described. If the authors would like to describe the correlation of SOX11 as a biomarker, then these data would be appropriate, however they would still need to explain why SOX11 is a better marker for tumor aggressiveness than say, TMB, or already established biomarkers.

Round 3

Reviewer 2 Report

In this revised version, authors improved the presented data. Also they tried to answer all reviewer comments. Therefore, I am accepting this work for publication in its present form.

Author Response

Thank you very much for your comments and professional advice. These opinions indeed help to improve academic rigor of our article.